# Decentralized Diagnosis: Privacy-Preserving Brain Tumor Classification with Federated Learning

Chaima Lhasnaoui
Computing Systems Engineering Laboratory
Cadi Ayyad University
Marrakesh, Morocco
c.lhasnaoui.ced@uca.ac.ma

Addi Ait-Mlouk
SAIL group, University of Skövde
Skövde, Sweden
addi.ait-mlouk@his.se

Tarik Agouti
Computing Systems Engineering Laboratory
Cadi Ayyad University
Marrakesh, Morocco
t.agouti@uca.ac.ma

Mohammed Sadgal
Computing Systems Engineering Laboratory
Cadi Ayyad University
Marrakesh, Morocco
sadgal@uca.ac.ma

## ABSTRACT

Brain tumors pose a significant global health challenge, driving ongoing research advancements in early detection methods. Artificial intelligence (AI) and deep learning (DL) techniques have shown great potential in this field, enabling the creation of highly accurate models for brain tumor identification from medical images. However, centralized approaches to these methods often raise critical concerns regarding patient data privacy and security. This paper presents a novel federated learning (FL) framework for brain tumor identification that effectively addresses these privacy concerns. FL enables collaborative model training across multiple institutions without the need for raw data sharing. Each participating institution trains the model locally on their Magnetic Resonance Imaging (MRI) datasets and only transmits model updates to a central server for secure aggregation. This iterative process results in a robust global model trained on a distributed dataset while preserving patient data confidentiality. The proposed FL model is evaluated using a dataset of 3,000 MRI images. Experimental results demonstrate the effectiveness of our approach, achieving a high accuracy rate of 96.88% for brain tumor identification. These findings suggest that FL provides a viable solution for privacy-preserving brain tumor identification, maintaining comparable performance to centralized models while ensuring the security of patient data.

## CCS CONCEPTS

• **Social and professional topics** → **Medical technologies**; • **Computing methodologies** → **Machine learning**; • **Computer systems organization** → **Neural networks**; • **Security and privacy** → *Privacy protections.*

## KEYWORDS

Brain Tumor, Federated learning, Classification, Data privacy, Deep learning, Medical imaging, Machine learning.

**ACM Reference Format:**
Chaima Lhasnaoui, Addi Ait-Mlouk, Tarik Agouti, and Mohammed Sadgal. 2024. Decentralized Diagnosis: Privacy-Preserving Brain Tumor Classification with Federated Learning. In *KDD-AIDSH 2024: Artificial Intelligence and Data Science for Healthcare: Bridging Data-Centric AI and People-Centric Healthcare, August 25–29, 2024, Barcelona, Spain.* ACM, New York, NY, USA, 6 pages. https://doi.org/10.1145/nnnnnnn.nnnnnnn

## 1 INTRODUCTION

Brain tumors pose a critical health challenge with a high mortality rate, making early detection essential for improving patient survival. These tumors are abnormal masses of cells that form in the brain and can be classified into primary tumors, which develop directly in the brain, and metastatic tumors, which spread to the brain from other parts of the body [19]. MRI is the most effective diagnostic tool for obtaining detailed images of the brain, aiding in the differentiation of tumors from normal brain tissues. However, while MRI is effective, it relies heavily on the interpretation of radiologists [1], which can introduce subjectivity. Additionally, the large volume of data generated can complicate the analysis. In response, AI, particularly DL technologies, has emerged as a powerful tool for the early detection of brain tumors, enhancing performance in various medical sectors [4]. Several studies have developed DL models capable of accurately and efficiently identifying brain tumors [1, 5, 11, 15, 20]. However, a significant challenge with these approaches is their centralized nature, which requires the sharing of data in a central repository. This centralization raises concerns about patient privacy and data security, which are critical in medical applications.

Confronted with the challenges of centralized deep learning, FL emerges as a promising solution to address data privacy and communication costs [6]. FL is an innovative approach to machine learning that allows multiple entities (clients) to collaborate in solving machine learning problems under the coordination of a central server. Unlike traditional centralized machine learning approaches, where patient MRI data is aggregated in a single location,

FL empowers individual institutions to train the model locally on their own datasets. This eliminates the need for raw data exchange or transfer [21], significantly mitigating privacy risks associated with centralized data storage. This decentralized approach not only mitigates privacy concerns but also reduces the risks associated with centralized data storage. By leveraging FL, it is possible to maintain high model performance while ensuring patient data security, thereby overcoming a key limitation of current AI and DL technologies in medical applications.

The remainder of this paper is organized as follows. Section 2 surveys related work. Section 3 outlines the methodology used and its implementation. Section 4 presents the results and engages in a discussion that compares the centralized and federated approaches, as well as a comparison with previous research. Section 5 provides a conclusion summarizing the main findings of this study.

## 2 RELATED WORK

Prior research has extensively explored DL algorithms for brain tumor identification. Tun Azshafarrah et al. [9] investigated the performance of VGG-16, ResNet-50, and AlexNet architectures on brain MRI images. Their findings suggest that AlexNet achieved superior accuracy, precision, and recall compared to the other models. Similarly, Francisco Javier et al. [8] proposed a multiscale CNN for brain tumor classification and segmentation, achieving high accuracy. Building upon these advancements, Naeem Ullah et al. [15] introduced TumorDetNet, a model based on an improved MobileNet architecture, demonstrating superior performance in brain tumor detection and classification.

In the context of federated learning, Moinul Islam et al. [14] applied CNN models and FL for the detection of brain tumors from MRI images. The researchers evaluated six pre-trained CNN models: VGG16, Inception V3, VGG19, ResNet50, Xception, and DenseNet121. Among these, VGG16, Inception V3, and DenseNet121 showed the best individual performance. To optimize these results, the researchers created an average model by combining VGG16, Inception V3, and DenseNet121, leveraging their complementary strengths. Similarly, Meenakshi Aggarwal et al. [12] focused on developing a collaborative transfer learning model for the multiclass classification of brain tumors. They trained and evaluated various CNN models, including six transfer learning models, using performance metrics such as precision, recall, and loss. Their study identified DenseNet201 as the best neural network for brain tumor classification. Furthermore, they implemented a FL approach to maintain the confidentiality of patient data while ensuring robust model performance.

Our study makes a significant contribution to the field of medical imaging and diagnostics by introducing a novel approach for brain tumor identification leveraging federated learning. We employed a CNN structure based on VGG16 to construct our model, and our findings demonstrate that our approach outperforms previous results. By conducting a comprehensive comparison with centralized methods, we highlight the superiority of our federated learning-based approach in terms of both accuracy and privacy preservation.

## 3 METHODS

The overall architecture of our proposed approach is shown in Figure 1. We employed a federated learning approach for brain tumor detection. This technique leverages a CNN architecture with a VGG16 backbone trained collaboratively across multiple institutions. The FL framework facilitates model training on local datasets at each institution (client) without compromising individual data privacy (Figure 1). Table 1 summarizes the configuration parameters used in this FL approach, including the number of communication rounds, training epochs per round, and the total number of participating clients. During each training round, clients update the local model's parameters and send only these updated weights to the central server, ensuring data privacy (Figure 1). Parallel training across clients maximizes learning speed and resource utilization. The server, implemented using the Flower library, orchestrates the FL process. It employs a custom function to aggregate accuracy metrics received from clients, calculating their average to obtain global metrics after each round. The Flower library's FedAvg strategy is then used to update the global model based on the received client updates. Hardware and software specifications for this study are summarized in Table 2.

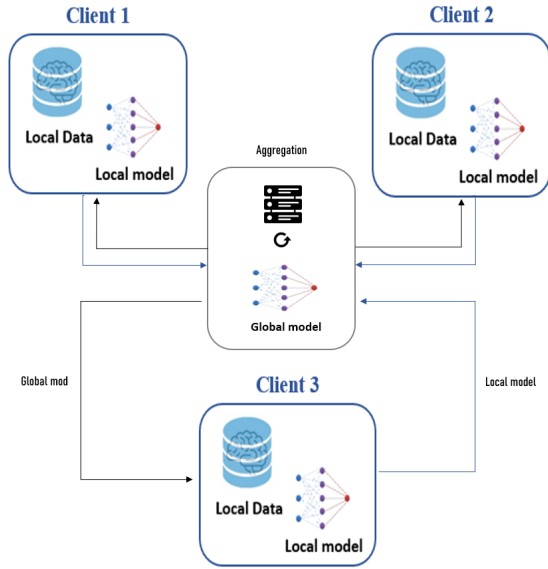

**Figure 1: Federated learning approach**

**Table 1: Federated Learning Configuration Parameters**

| Rounds | Nbr of Client | Nbr of Epochs |
|--------|---------------|---------------|
| 10     | 3             | 10            |

**Table 2: Hardware and Software Specification**

| Item | Detail |
|------|--------|
| Deep learning framework | TensorFlow 2.15.0 |
| Federated learning framework | flower |
| GPU | Intel (R) UHD Graphics 620 |
| CPU | Intel Core i7-8650U |
| RAM | 16 Go |
| Operating system | Windows 11 |

## 3.1 Data

For this study, we utilized a dataset of 3,000 brain MRI images obtained from Kaggle [7]. The dataset is specifically designed for brain tumor classification and comprises 1,500 MRI images labeled as "Yes" (tumorous) and 1,500 labeled as "No" (non-tumorous) to ensure balanced representation of both classes. An illustrative example of these images is provided in Figure 2. Given the importance of handling limited datasets effectively, we integrated various data augmentation techniques into our analysis. Data augmentation is a pivotal strategy in machine learning, as it allows for the expansion and diversification of training data by generating new instances from existing ones through transformations like flipping and rotation while retaining the original labels. This approach helps alleviate overfitting concerns and enhances the model's ability to generalize to unseen data during inference. Notably, studies by Cossio et al. [10] and Wang et al. [3] underscore the substantial impact of data augmentation on deep learning model robustness. Our utilization of diverse data augmentation techniques, as depicted in Figure 3, further strengthens the reliability and generalizability of our findings.

## 3.2 CNN Architectures

Our approach leverages the VGG16 architecture, a widely adopted CNN for image classification tasks [9, 14, 16]. VGG16's ability to extract complex features from high-dimensional images makes it well-suited for the detailed nature of medical images. We utilize the VGG16 base, excluding its fully connected layers, to offer greater flexibility in adapting to brain MRI images while retaining pre-learned features. Extracted features are then flattened for classification. A dense layer with a ReLU activation function introduces the non-linearity required for learning distinctive brain tumor features. To mitigate overfitting, we incorporate a dropout layer with a 50% dropout rate, randomly deactivating half of the neurons during training [17]. Finally, the output layer employs a sigmoid activation function to generate the probability of a tumor being present in an image, facilitating binary classification. Our model is compiled with the Adam optimizer and a binary crossentropy loss function. Details regarding the evaluation metrics used to assess model performance in a medical context, are provided in Section 4.

## 4 RESULTS AND DISCUSSION

### 4.1 Metrics

We evaluated our model's brain tumor detection effectiveness using precision 1, recall 2, accuracy 3, and F1-score 4. Precision reflects the proportion of true positives (TP) among all positive predictions (TP + FP). In other words, it measures the accuracy of the model's positive classifications (tumor identified). Recall, on the other hand, focuses on the true positive rate (TPR), representing the percentage of actual tumor cases (TP) correctly identified by the model out of all actual tumors (TP + FN). It highlights the model's ability to capture true tumor cases. Accuracy, a more general metric, encompasses both correctly classified tumors and non-tumors, providing a combined measure of performance (correctly classified cases / total cases). Finally, the F1-score offers a harmonic mean of precision and recall, balancing these two aspects of model performance.

$$Precision = \frac{TP}{TP + FP} \qquad (1)$$

$$Recall = \frac{TP}{TP + FN} \qquad (2)$$

$$Accuracy = \frac{TP + TN}{TP + TN + FP + FN} \qquad (3)$$

$$F1\ score = 2 \times \frac{(Precision \times Recall)}{(Precision + Recall)} \qquad (4)$$

### 4.2 Centralized approach Evaluation

We employed 5-fold cross-validation for centralized model training [18]. This technique iteratively divides the data into five folds, using four folds for training and one for validation in each epoch. This process ensures all data segments participate in both training and validation throughout the training process (batch size: 32, epochs: 30). The model achieved an accuracy of 97% after training. To comprehensively evaluate performance, we calculated precision, recall, and accuracy by averaging the results obtained across all five folds of the cross-validation process. These performances are visualized in Figure 4.

### 4.3 Federated learning Evaluation

Following successful validation of the CNN model in the centralized setting, we leveraged it to establish the global model within the FL environment. We divided the data among the three clients, each having 1000 MRI scans: 500 with tumors and 500 without. Their datasets were split into 80% for training and 20% for validation. To train the local models at each client, we employed 10 epochs. For model aggregation, the central server utilizes FedAvg [2], which averages the local model updates received from all clients. This aggregated update is then added to the global model, fostering collaborative learning without data exchange. The FL approach was evaluated over 10 communication rounds, measuring the global model's accuracy at each iteration (Table 4). Our results demonstrate that the FL model achieves high accuracy in brain tumor identification while preserving data privacy. As shown in Figure 5 and Figure 6, performance metrics improve with increasing communication rounds.

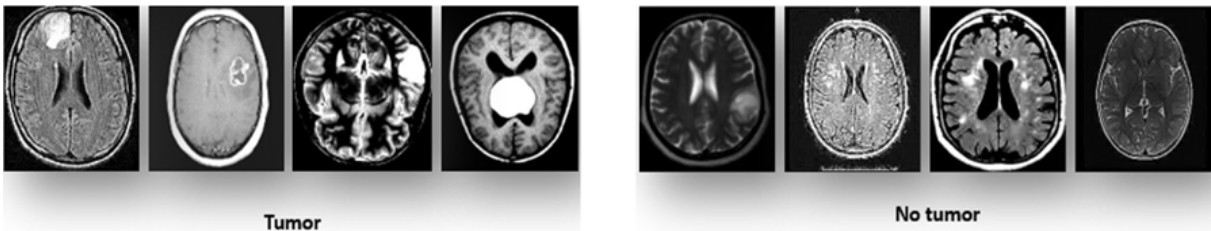

Figure 2: Example Image of Brain Tumor Dataset.

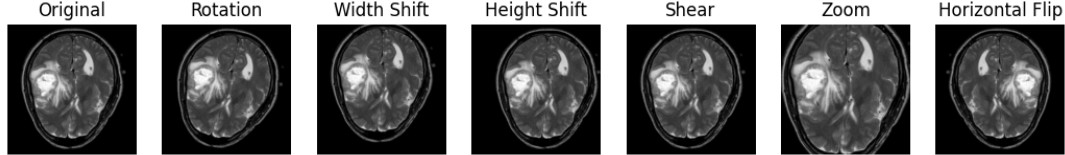

Figure 3: Augmented Images with Various Transformations Applied to the Original Image.

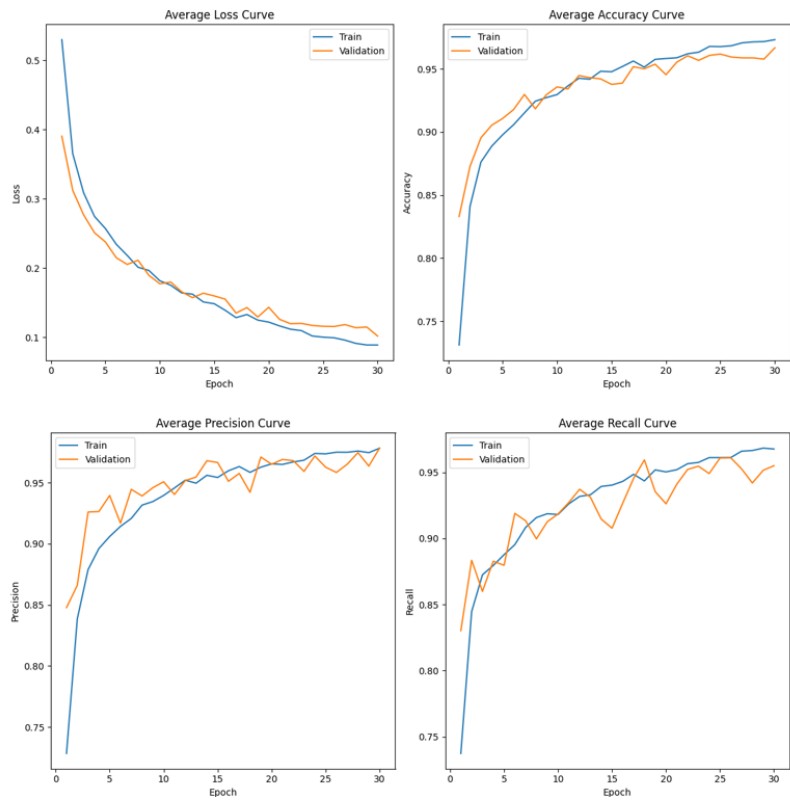

Figure 4: CNN Model Training and Validation. This figure depicts the performance of the CNN model on the training and validation sets.

## 4.4 Discussion

Our proposed federated learning model achieves competitive performance when compared to existing centralized approaches for brain tumor identification. This section provides a comparative analysis of relevant techniques, highlighting the trade-offs between accuracy and data privacy. Several studies have explored DL models for brain tumor classification using MRI images with promising results. Toğaçara et al. [13] propose BrainMRNet, a novel MRI-based approach utilizing residual blocks, hypercolumn techniques, and attention modules, achieving an accuracy of 96.05%. Similarly,

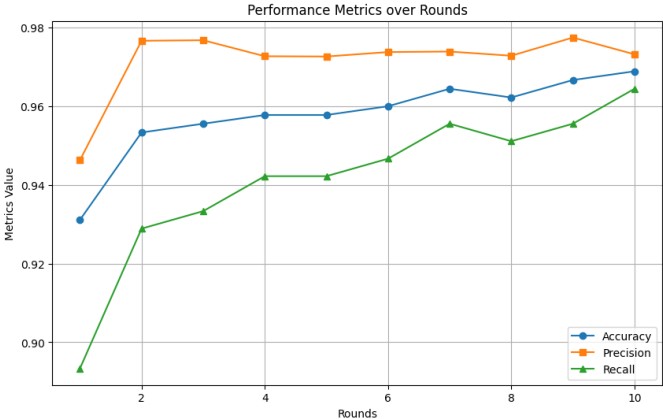

**Figure 5: Performance Metrics of the Federated Learning Model across Communication Rounds. This figure depicts how metrics such as accuracy, precision, recall, and F1-score evolve as the number of communication rounds increases in the federated learning process.**

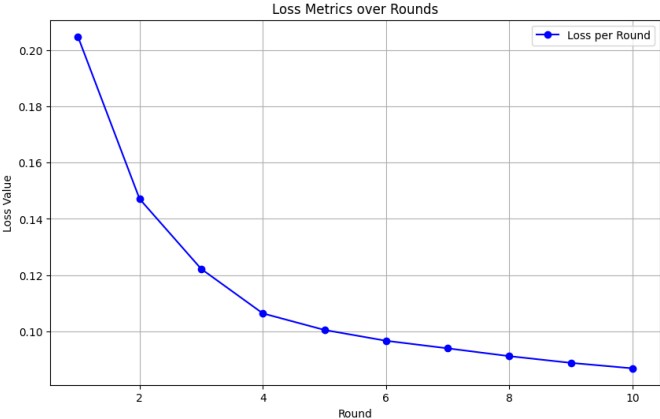

**Figure 6: Convergence of Federated Learning Loss. This figure illustrates the convergence of the loss function in the federated learning model, signifying a decrease in training error over communication rounds.**

Azshafarrah et al. [9] report that AlexNet yields the highest accuracy (96.10%) among VGG-16 and ResNet-50 architectures. While these approaches demonstrate high accuracy, they often rely on centralized data storage, raising privacy concerns. Table 3 summarizes these comparisons.

Moinul Islam et al. [14] also employed FL for brain tumor classification using the same dataset, achieving an accuracy of 93.22%. Their accuracy dropped to 91.05% with a different dataset. Our FL model surpasses their results with an accuracy of 96.88%, demonstrating the effectiveness of our proposed approach. These findings indicate that FL while preserving data privacy, can achieve performance comparable to centralized models. Our centralized approach achieved 97% accuracy using cross-validation, while our

**Table 3: Comparison of Brain Tumor Detection Methods. This table summarizes the performance and characteristics of various deep learning approaches for brain tumor detection, including our proposed federated learning model.**

| Paper | Data size | Approach Used | Accuracy |
|---|---|---|---|
| [9] | 253 images | VGG-16 | 94.16% |
| | | ResNet-50 | 91.56% |
| | | AlexNet | 96.10% |
| [13] | 253 images | BrainMRNet | 96.05% |
| [14] | 2309 images | CNN + FL | 91.05% |
| | 3000 images | CNN + FL | 93.22% |
| **Our Model** | **3000 images** | **CNN + FL** | **96.88%** |

**Table 4: Comparison of Centralized and Federated Learning Approaches for Brain Tumor Detection. This table summarizes the performance of both approaches on various metrics (accuracy, precision, recall, F1-score).**

| Model | Accuracy | Precision | Recall | F1 score |
|---|---|---|---|---|
| Centralized | 0.97 | 0.98 | 0.96 | 0.97 |
| Federated | 0.96 | 0.97 | 0.96 | 0.96 |

FL approach attained 96.88% without cross-validation. This minimal difference highlights the potential of FL for secure collaborative learning in medical diagnosis. Table 4 provides a more detailed comparison of these approaches using various metrics. In conclusion, our FL approach offers a compelling alternative, preserving data privacy while maintaining competitive performance. This emphasizes the potential of FL for collaborative medical diagnosis without compromising sensitive patient information.

## 5 CONCLUSION

This study investigated the application of FL for brain tumor identification, addressing the challenge of limited and siloed medical datasets due to privacy concerns and regulations. We proposed a VGG16-based CNN model trained within a FL framework, ensuring data privacy while facilitating collaborative learning across distributed healthcare institutions. To the best of our knowledge, our FL approach achieves state-of-the-art performance in brain tumor identification. Our model's accuracy of 96.88% is highly competitive with the centralized approach (97.0%), demonstrating the potential of FL to maintain high accuracy while preserving data privacy. This paves the way for secure and collaborative medical diagnosis in the domain of brain tumors. Our research opens exciting avenues for further exploration. We aim to expand our approach to incorporate multi-class classification, enabling the identification of various brain tumor types. Additionally, we will focus on enhancing the privacy-preserving mechanisms for resource-constrained edge devices, enabling secure and collaborative diagnosis at the edge.

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
