# OpenReview forum: "Decentralized Diagnosis: Privacy-Preserving Brain Tumor Classification with Federated Learning"
_KDD.org/2024/Workshop/AIDSH — KDD-AIDSH 2024 Poster_

### Official Review · Reviewer_e9NM · 2024-06-14
**Review for Paper #38**

**Rating:** 3
**Confidence:** 4

**Review:**

## Summary

This paper utilizes a federated learning (FL) framework for brain tumor identification from MRI images to address patient data privacy concerns associated with centralized approaches. The proposed method involves training a CNN model locally at multiple institutions and aggregating model updates on a central server without sharing raw data. Experimental results on a dataset of 3,000 MRI images demonstrate an accuracy rate of 96.88%.

## Pros

- Sound FL methodology design and the large volume of utilized data samples.

## Cons

- Only one dataset is evaluated, which may not be representative of the broader variability in medical imaging datasets.
- Is the performance comparable? The results of multiple baseline methods in Table 3 use different data sizes. Do they share the same dataset? Did you really reproduce the results with a fair comparison?
- The performance should include mean and standard deviation results.
- The implementation details and presentation are not clear. For example, what specific i7 CPU? Typo: 16GB. The Horizontal Flip does not exhibit flip results compared to the original image, as well as the Fill Mode; no clear difference is shown.
- The introduction of metrics is redundant. The AUROC and AUPRC scores should be provided.

## Additional Note

The paper disobeys the double-blind rule. It includes the authors and their affiliations.

---

### Official Review · Reviewer_XPXT · 2024-06-18
**review：Decentralized Diagnosis: Privacy-Preserving Brain Tumor Classification with Federated Learning**

**Rating:** 6
**Confidence:** 4

**Review:**

This manuscript declaims that a novel federated learning (FL) framework is proposed for brain tumor identification that effectively addresses these privacy concerns. The proposed FL model is evaluated using a dataset of 3,000 MRI images. Experimental results demonstrate the effectiveness of our approach, achieving a high accuracy rate of 96.88% for the brain tumor identification task. But the novelty and contribution of this manuscript are limited, it only combines VGG16 and the concept of FL for brain tumor classification.
1. The experimental results only indicate that there are three clients in the Federated Learning system.
2. It is crucial to include more credible analytical results beyond showcasing training curves. For instance, adding analyses such as feature visualization can enhance the credibility and contribution of the work.
3. To enhance the novelty of the methods, consider introducing innovative approaches. Additionally, analyze and highlight the contribution of clinical medical imaging. Incorporating comparative experiments with state-of-the-art methods can also strengthen the work.

---

### Decision · Program_Chairs · 2024-06-28

Accept (Poster)